# Vancomycin nephrotoxicity: A comprehensive clinico-pathological study

**Rajesh Nachiappa Ganesh**[1,2]*, **Angelina Edwards**[3], **Ziad El Zaatari**[2], **Lillian Gaber**[2], **Roberto Barrios**[2], **Luan D. Truong**[2]

**1** Department of Pathology, Jawaharlal Institute of Postgraduate Medical Education and Research (JIPMER), Puducherry, India, **2** Department of Pathology and Genomic Medicine, Houston Methodist Hospital, Houston, Texas, United States of America, **3** Division of Nephrology, Department of Medicine, Houston Methodist Hospital, Houston, Texas, United States of America

* drngrajesh@yahoo.co.in, rnganesh@houstonmethodist.org

**Data Availability Statement:** All available data are provided with manuscript

**Funding:** The author(s) received no specific funding for this work.

## Abstract

### Introduction

Vancomycin, a commonly prescribed antibiotic particularly in the setting of multi-drug resistant infections, is limited by its nephrotoxicity. Despite its common occurrence, much remains unknown on the clinicopathologic profile as well as the pathogenesis of vancomycin nephrotoxicity. Clinical studies included patients often with severe comorbidities and concomitant polypharmacy confounding the causal pathogenesis. Animal models cannot recapitulate this complex clinical situation. Kidney biopsy was not commonly performed.

### Methods

To address this limitation, we studied 36 patients who had renal biopsies for acute kidney injury (AKI) for suspicion of vancomycin nephrotoxicity. Detailed renal biopsy evaluation, meticulous evaluation of clinical profiles, and up-to-date follow-up allowed for a diagnostic categorization of vancomycin nephrotoxicity (VNT) in 25 patients and absence of vancomycin nephrotoxicity (NO-VNT) in 11 patients. For careful comparison of these two groups, we proceeded to compile a clinicopathologic and morphologic profiles characteristic for each group.

### Results

Patients with VNT had a characteristic clinical profile including a common clinical background, a high serum trough level of vancomycin, a rapidly developed and severe acute kidney injury, and a recovery of renal function often shortly after discontinuation of vancomycin. This clinical course was correlated with characteristic renal biopsy findings including acute tubulointerstitial nephritis of allergic type, frequent granulomatous inflammation, concomitant and pronounced acute tubular necrosis of nephrotoxic type, and vancomycin casts, in the absence of significant tubular atrophy and interstitial fibrosis. This clinico-pathologic profile was different from that of patients with NO-VNT, highlighting its role in the diagnosis, management and pathogenetic exploration of vancomycin nephrotoxicity

**Competing interests:** The authors declare that no competing interests exist.

## Conclusion

Vancomycin nephrotoxicity has a distinctive morphologic and clinical profile, which should facilitate diagnosis, guide treatment and prognostication, and confer pathogenetic insights.

## Introduction

Staphylococcal infection resistant to conventional antibiotics including methicillin is increasingly frequent and represents one of the most severe types of infection. Consensus guidelines from the Infectious Diseases Society of America indicate that vancomycin is the drug of choice in this situation. In fact, vancomycin is the most frequently prescribed antibiotic in the hospital setting for up to 35% of hospitalized patients with infection [1].

One of the more frequent side effects of vancomycin is nephrotoxicity (VNT), which is reported in up to 25% of patients and manifests clinically as acute kidney injury (AKI) developing *de novo* or from a background of chronic kidney injury (CKI) [2].

Despite the profound significance of VNT, much remains unknown of the various clinical presentations, pathologic findings, and pathogenesis of this condition. Limitations on the pertinent literature may account for this deficiency.

Most studies on VNT, although involving a large number of patients, represent retrospective clinical evaluation, lacking a comparative group or renal biopsy support, and are typically biased by concomitant confounding factors. In these studies, the majority patients had witnessed acute conditions, which can cause AKI by themselves including septicemia, severe localized infection, electrolyte imbalance, intravenous contrast agents, or concomitant potentially nephrotoxic antibiotics or other medications. In additions, many also had pre-existing chronic comorbidities such as diabetes mellitus and hypertension, which by themselves may contribute to an element of AKI on CKI, independent from vancomycin administration. Thus, the diagnosis of VNT may not be certain in many of them. In fact, Sinha *et al* in an in-depth meta-analysis suggested that in patients treated with vancomycin who develop AKI, only 59% of cases can be attributed to VNT [3, 4].

Kidney biopsies from the affected patients might have alleviated this conundrum. Yet, very few kidney biopsies were performed in these patients, reflecting an often-rapid clinical improvement after a stop of vancomycin, a rather typical feature of VNT, negating a need for kidney biopsy, or a reluctance to perform an invasive procedure on an already quite sick patient. Hence, the literature on kidney biopsy in the context of VNT includes only few individual case reports, describing acute tubular necrosis (ATN) or tubulo-interstitial nephritis (TIN), without an in-depth morphologic characterization or exploring the pathogenetic significance of the renal biopsy findings.

Study of VNT on experimental animals is eminently feasible and several of them have been completed. Yet, these studies are done only in healthy animals totally lacking the complex clinical context encountered in human VNT. The reported renal changes are quite mild, and are often not akin to the human findings, casting doubt on the validity of these models to study the pathogenesis of VNT.

Recent insight into the pathogenesis of VNT was offered in an elegant study by Luque and coworkers in 2017, which meticulously documents tubular casts containing vancomycin (vancomycin casts) in mice injected with vancomycin, and further identified them in few patients with VNT [5]. This was followed by a comprehensive study in 2021 on kidney biopsies from our registry with a possible diagnosis of VNT, focusing only on vancomycin casts. This study

confirms the presence of vancomycin casts in most but not all these biopsies and proceeds to describe its morphology in meticulous detail with light microscopy examination, immunohistochemistry, as well as electron microscopy [6]. Yet, in both studies, aside from an exhaustive examination of vancomycin casts, other important attributes of VNT such as the overall renal biopsy findings, clinical manifestation, or disease course, are not addressed. Thus, the clinical features of VNT as a disease, its pathogenesis including structural basis, and the role of vancomycin casts in this process remain unsettled.

Against this background of limited knowledge, a better appreciation of the clinicopathologic profile of VNT would certainly improve patient care, including risk assessment, management, and prognostication. Furthermore, renal biopsy may be performed to help identify the cause of AKI, which is essential for the crucial decision to continue the vancomycin or to withdraw it at the risk of progressive infection. Knowledge on the renal biopsy changes caused by vancomycin would not only help elucidate the pathogenesis of VNT, but also instrumental for clinical decision and management [7].

The current study aims to comprehensively evaluate the clinical manifestations and the renal changes of vancomycin nephrotoxicity. This comprehensive clinicopathologic study, harnessing the power of renal biopsy and a careful clinical documentation and follow-up, in addition to the inclusion of a comparative group, aims to address some of the limitations of previous studies and hopefully provide additional insights into VNT and its pathogenesis.

## Materials and methods

The renal biopsy registry from the Houston Methodist Hospital System and Baylor College of Medicine within the 2010–2021 period was reviewed to identify the biopsies performed in patients who were treated with vancomycin around the time of biopsy. Fifty biopsies were identified among a total of 5598 biopsies performed during this period. The indication for each of these biopsies was AKI, with or without other manifestations including proteinuria and hematuria. Our premise is that although all biopsied patients were treated with vancomycin, AKI was caused by vancomycin in only some of them, thus a group of patients with vancomycin nephrotoxicity (VNT) and another group without VNT (NO-VNT) can be constructed for comparison, which may help to define the distinctive clinical and morphologic profiles of VNT. For this goal, clinical information including adequate follow-up was collected. Follow-up records was not available in 14 cases, and these cases were not studied further. Among the remaining 36 cases, careful retrospective examination of both clinical and renal biopsy findings detailed below helped define 25 cases of VNT and 11 cases of NO-VNT for final study. It is noted that our previous study focusing on vancomycin casts included 37 biopsies, but only 20 of them were included in the current study, reflecting stringent inclusion criteria. The biopsy was performed to evaluate the cause of acute kidney injury in patients who were treated with vancomycin. The time interval between the onset of vancomycin treatment and renal biopsy ranged from 2–69 days, with a median of 6 days. Baseline serum creatine is defined as the lowest level before the administration of vancomycin. This value is either recorded during a previous hospital admission for reasons unrelated to the current acute kidney, or during the current admission, before the development of acute kidney injury.

Vancomycin is frequently used to treat infection. Acute kidney injury may develop in association with this treatment. There is a need to determine whether the acute kidney injury is due to vancomycin requiring cessation of this treatment or it is unrelated to vancomycin so that vancomycin can be continued. This dilemma not infrequently leads to divergent opinions of the treating nephrologist and infection specialist. The clinical manifestations and the renal biopsy findings, which may be distinctive for vancomycin nephrotoxicity, may be helpful for

this essential differential diagnoses. However, the pertinent current literature is very limited. In an effort to address this matter, we have attempted to classify the patients into a group with vancomycin nephrotoxicity and those without vancomycin nephrotoxicity for a comprehensive comparative evaluation. This decision may be problematic in some cases, especially at the time of patient care, but is significantly facilitated in the current retrospective review. Accordingly, the patients with VNT displayed a distinctive clinical profile as detailed in the results, Table 1 and discussion. In contrast, the reason for the diagnosis of NO-VNT included 1) AKI long *after* discontinuation of vancomycin, with a normal serum creatinine at time of this discontinuation; 2) treatment with oral vancomycin only; 3) AKI developing before the start of vancomycin; 4) very slow recovery of renal function after discontinuation of vancomycin; 5) recovery of renal function when still on vancomycin; and 6) several other episodes of vancomycin treatment not associated with AKI. These findings were not recognized in the VNT group. Furthermore, as detailed in Table 1, more than one reason was noted from the composite criteria for each of the NO-VNT patients. The renal biopsy findings also helps to support the diagnosis of NO-VNT. Although renal biopsy changes due to vancomycin nephrotoxicity have not been well defined in literature, over the course of study, a characteristic and quite uniform profile of renal biopsy changes were identified (see below). These findings were not seen in any of the cases of NO-VNT.

For each kidney biopsy, light microscopic (LM) examination was proceeded with hematoxylin and eosin (H & E), periodic acid-Schiff (PAS), silver methenamine and Masson trichrome (trichome) stains. Immunofluorescence (IF) examination was performed for IgG, IgA, IgM, C3, C1q, C4, kappa and lambda light chains. Immunohistochemical staining for myoglobin and Mib-1, a cell proliferation marker, was done in each biopsy. Immunohistochemical stain was done to localize vancomycin and uromodulin in 27 cases. In 5 cases, immunohistochemical staining for the RCC Marker (a marker for proximal tubule) and kidney-specific cadherin (a marker for Henle loop and collecting duct) was done, in conjunction with the identified vancomycin casts for a precise localization of these casts along the nephron. Electron microscopy (EM) was done for each biopsy, with special attention to the tubulointerstitial changes and in-depth morphologic characterization of the tubular casts. The kidney biopsies were comprehensively evaluated for changes that may be due to VNT, with special attention to those that may be pertinent to VNT including ATN features, interstitial fibrosis/tubular atrophy (IFTA), tubulointerstitial nephritis (TIN) including distribution, density, and nature. A semi-quantitative scoring was adopted as followed: For IFTA and TIN: 0 = 0–10%, 1 = 10–25%, 2 = 25–50%, and 3 = >50% of renal tissue area, focusing on cortex; for ATN: 0 = absent, 1 = few (<25%) dilated tubular profiles with mild tubular injury, 2 = more (25–50%) tubules with dilatation and moderate tubular cell injury, and 3 = many (>50%) tubular profiles affected with severe acute tubular cell injury; for vancomycin casts: 0 = absent, 1 = very occasional casts in 1–2 tubules, 2 = presence of casts in 3–5 tubules, and 3 = many casts in > 5 tubules; for arteriosclerosis: 0 = normal, 1 = intimal fibrosis occupying > 25% of original vascular lumen, 2 = 25–50%, and 3 = > 50%, for diabetic nephropathy (DN): I = mild nonspecific LM changes with thickened glomerular basement membrane by EM, IIa = mild mesangial expansion, IIb = marked mesangial expansion, III = Mesangial nodules, and IV = advanced sclerosis.

Institute review board approval was obtained (MOD0006414) and medical records were reviewed. The clinical and research activities being reported are consistent with the principles of the Declaration of Istanbul. Waiver was obtained from informed consent, as it was a retrospective study, without direct patient contact or intervention.

Comprehensive data abstraction from medical record was made, including comorbid conditions with known predisposition to renal injury, potential precipitators of AKI such as other

**Table 1. Tabulated summary of clinical features of patients with and without vancomycin nephrotoxicity.**

| Case # | Gender | Age (yrs.) | Background Diseases | Possible Causes of AKI, other than VNT and Infection | Primary infection | Culture report | Vancomycin Dose | Duration (days) | Trough Blood Levels of VAN (Microgram/dL) | Other Antibiotics | Creatinine (mg/dL) Baseline | Creatinine (mg/dL) Peak or at Bx | Proteinuria at Bx | Interval between Van Start and AKI (days) | Interval between AKI and Bx (days) | Interval between stopping Van and baseline Serum Creatinine or Last Follow-up Serum Creatinine (days) | Diagnostic Considerations |
|---|---|---|---|---|---|---|---|---|---|---|---|---|---|---|---|---|---|
| | | | | | | | | | **VANCOMYCIN NEPHROTOXICITY** | | | | | | | | |
| 1 | F | 66 | DM, HT, Athero | Bypass surgery, drugs | Sternal wound, empyema, pneumonia | E faecalis | 1 g/q day | 30 | 32 | Ampicillin | 1.1 | 8.9 | Trace | 20 | 26 | Back to baseline in 36 days | Typical clinical course and renal Bx |
| 2 | F | 23 | DM, HT, Athero | Drugs | Leg cellulitis | Streptococcus | 1 g/q day | 15 | 25 | None | 0.7 | 6 | Trace | 9 | 1 | Back to baseline in 15 days | Typical clinical course and renal Bx |
| 3 | M | 24 | HIV | HIV, septicemia | Knee joint | Negative | 1 g/q day | 10 | 37 | Bactrim, ceftriaxone | 1.1 | 4.7 | 1+ | 2 | 4 | Back to baseline in 16 days | Typical clinical course and renal Bx |
| 4 | F | 53 | DM, HTN, HL, Athero | Drugs, radiocontrast agents | Right thigh abscess | Blood neg, wound Serratia | 1 g/day | 6 | 18 | Azithromycin, cefepime | Unknown (4.4 at presentation with infection) | 7.3 | 2+ | 2 | 15 | Back to baseline in 10 days | Typical clinical course and renal Bx |
| 5 | F | 55 | HTN, HL, morbid obesity | Drugs, radiocontrast agents | Pneumonia | Negative | 1 g / q 12 hrs | 7 | 15 | Piperacillin/ tazobactam | 0.7 | 11 | 1+ | 7 | 6 | Back to baseline in 7 days | Typical clinical course and renal Bx |
| 6 | F | 51 | Crohn disease | Drugs | Tendon abscess | Negative | 1q 8 hrs first day, then 1.5g/q 12 hrs | 15 | 51 | Ceftriaxone | 0.5 | 4.9 | 0.3 | 15 | 10 | Back to baseline in 20 days | Typical clinical course and renal Bx |
| 7 | M | 53 | DM, HTN, HL | Drugs | Cellulitis, foot | SA | 2g/d day | 3 | 22 | Ofloxacin, linezolid, cefepime | 0.8 | 8.9 | 266mg/ day | 2 | 2 | Back to baseline in 11 days | Typical clinical course and renal Bx |
| 8 | M | 38 | Obesity | Radio-contrast agents | Perianal abscess | Bacteroid, E coli, streptococcus | 1 g/ q day | 2 | Not done | Piperacillin/ tazobactam | 1.1 | 13 | 500mg/day | 3 | 1 | Back to baseline in 27 days | Typical clinical course and renal Bx |
| 9 | M | 38 | DM, HTN, morbid obesity, gout | Radio-contrast agents | Neck cellulitis, septicemia | Negative | 2g one dose, then 1.5 g/ 12 hrs | 4 | 18 | Linezolid, Piperacillin/ tazobactam | 1.2 | 4.4 | trace | 4 | 6 | Back to baseline in 25 days | Typical clinical course and renal Bx |
| 10 | M | 46 | DM, HTN, HL | Dehydration, radio-contrast agents | Osteomyelitis, toe | SA wound, negative blood | 1g/q 8 hrs | 3 | 18 | Piperacillin / Tazobactam | 0.8 | 7.6 | 1+ | 3 | 9 | Back to baseline in 28 days | Typical clinical course and renal Bx |
| 11 | M | 60 | HTN, CHF, sleep apnea | CHF, drugs | Pneumonia | MRSA | 1 g/day | 12 | 32 | Piperacillin/ tazobactam | 1.1 | 4.4 | P/C 0.7 | 6 | 46* | ** = Serum creat back to baseline 10 days after discontinuation of van, renal biopsy done 46 days later for heavy proteinuria; Van = Vancomycin; VNT = Vancomycin nephrotoxicity. | Typical clinical course |
| 12 | M | 53 | DM, Athero, morbid obesity | Dehydration, septicemia | Gangrenous ulcer, osteomyelitis, foot | Negative | 1500mgIV/ q12 hrs | 6 | 86 | Piperacillin/ tazobactam | 0.8 | 4.6 | 515 mg/day | 3 | 6 | Back to baseline in 60 days | Typical renal Bx |
| 13 | F | 49 | DM, HTN, morbid obesity, gout | Dehydration, drugs | Wound both legs | MRSA | 1.25 g/12 hrs | 69 | 43. | Cefepime | 0.5 | 5.6 | Trace | 69 | 6 | Cret 5.6 down to 3 in 13 days. Lost follow-up | Typical renal Bx |
| 14 | F | 60 | Diverticulosis | Radiocontrast agents, dehydration | Pneumonia | Negative | 1.5g/q 12 hrs | 4 | 37 | Cefepime Piperacillin/ tazobactam | 0.5 | 4.6 | 1+ | 4 | 1 | Cret 4.6 down to 3.2 in 6 days. Cret 0.8 at 2 years | Typical clinical course and renal Bx |
| 15 | F | 55 | DM, HTN, morbid obesity, diverticulosis | Dehydration, bleeding | Abscesses, abdomen, chest, scalp | MRSA | 1.75q/ 12 hrs | 6 | 32 | Piperacillin/ tazobactam | 1 | 8 | P/C 0.6 | 2 | 9 | Back to baseline in 31 days | Typical clinical course and renal Bx |
| 16 | F | 64 | DM, HTN, morbid obesity, heart disease | CHF, drugs | Necrotizing fasciitis, abdomen | Actinomyces | 2g one dose, then 1g/q 12 hrs | 5 | 28 | Piperacillin/ tazobactam, meropenem, clindamycin | 1.2 | 4 | 1+ | 2 | 16 | Back to baseline in 36 days | Typical clinical course and renal Bx |

*(Continued)*

**Table 1.** (Continued)

| Case # | Gender | Age (yrs.) | Background Diseases | Possible Causes of AKI, other than VNT and Infection | Primary infection | Culture report | Vancomycin Dose | Duration (days) | Trough Blood Levels of VAN (Microgram/ dL) | Other Antibiotics | Creatinine (mg/dL) Baseline | Creatinine (mg/dL) Peak or at Bx | Proteinuria at Bx | Interval between Van Start and AKI (days) | Interval between AKI and Bx (days) | Interval between stopping Van and baseline Serum Creatinine or Last Follow-up Serum Creatinine (days) | Diagnostic Considerations |
|---|---|---|---|---|---|---|---|---|---|---|---|---|---|---|---|---|---|
| 17 | M | 71 | HTN, HL, heart disease, hypothyroism | Drugs, hemodynamic | Knee prosthesis | MRSA | 1.25g/ q12 hrs | 13 | 27 | Cefepime, rifampin | 1.2 | 4.8 | 1 g/day | 6 | 11 | Back to baseline in 16 days | Typical clinical course and renal Bx |
| 18 | M | 26 | HTN, morbid obesity | Drugs | Rectal abscess | Negative | 1.75/q 12 hrs | 2 | 8.5 | Piperacillin/ tazobactam | 1 | 6.7 | 2+ | 3 | 3 | Cret from 6.7 down to 3.1 in 4 days. Cret 1 at 1 year | Typical clinical course and renal Bx |
| 19 | F | 78 | HTN, chronic obstructive lung disease, uterine carcinoma | Septicemia, bleeding, drugs, advanced carcinoma | Infected hematoma | MRSA | 0.75 q/12 hrs | 8 | 16 | Meropenem, micafungin | 0.5 | 1.92 | P/C 0.9 | 7 | 2 | Cret from 1.92 down to 1.43 in 27 days. Cret 1.2 at 18 months | Typical clinical course and renal Bx |
| 20 | M | 38 | Morbid obesity | Major surgery, drugs, radiocontrast agents | Abscess, thigh | MRSA | 1.25g/q 12 hrs | 5 | 21.5 | Piperacillin/ tazobactam ciprofloxacin | 0.9 | 9.7 | None | 3 | 5 | Cret 9.7 down to 6.2 in 3 days. Cret 1 at 2 years | Typical clinical course and renal Bx |
| 21 | M | 57 | HTN, DM, asthma | Drugs, septicemia, urine obstruction | Urinary tract, septicemia | MRSA | 1.25 g/q 12 hrs | 2 | Not done | Nafcillin | Unknown (2.2 at presentation with infection) | 7.3 | None | 2 | 4 | Cret 7.3 down to 5 in 4 days. Lost follow-up | Typical clinical course and renal Bx |
| 22 | F | 68 | HTN, HL, Athero | CHF, drugs | Hip prosthesis | Negative | 1g/q 12 hrs | 10 | 57 | Cefepime | 0.5 | 5.8 | None | 10 | 20 | Back to baseline in 97 days | Typical renal Bx |
| 23 | M | 45 | HTN, DM, obesity | Drugs | Foot osteomyelitis | Negative | 1.5g/q 12 hrs | 5 | 18 | Piperacillin/ tazobactam, ceftriaxone | 0.9 | 4.7 | 1+ | 2 | 7 | Cret 4.7 down to 3 in 1 day. Cret 0.8 in 7 months | Typical clinical course and renal Bx |
| 24 | M | 61 | DM, HTN, Athero | Drugs, dehydration | Foot abscess | MRSA | 1.5g/ q12 hrs | 6 | 23.5 | Piperacillin/ tazobactam | 0.7 | 8.1 | P/C 0.4 | 3 | 6 | Cret 8.1 down to 4 in 9 days. Cret 0.7 at 6 months | Typical clinical course and renal Bx |
| 25 | M | 46 | DM, HTN, HL | Radiocontrast | Foot abscess | Streptococcus | 1.5g/q12 hrs | 7 | 43 | Piperacillin/ tazobactam | 0.9 | 8.9 | None | 2 | 6 | Back to baseline in 24 days | Typical clinical course and renal Bx |
| **NO VANCOMYCIN NEPHROTOXICITY** | | | | | | | | | | | | | | | | | |
| 26 | F | 55 | DM, Athero, obesity | Drugs | Cellulitis, foot | MRSA | 1g/q day | 12 | 40 | None | 1.1 | 3.2 | 1+ | 5 | 1 | Not recovered, cret 4 at 1 year | Not recovered. Renal Bx no VNT changes |
| 27 | M | 59 | DM, HTN | Dehydration, drugs, radiocontrast agents | Skin abscess, osteomyelitis, elbow | Unknown | Oral only for C difficile colitis | 10 | NA | Daptomycin, Cefepime | 2 | 10 | 1+ | Only oral Van | 10 | Not recovered. Cret 3.7 at 1 year | AKI before Van; only oral Van. Renal Bx no VNTchanges |
| 28 | M | 72 | HTN | None | Seroma, hip prosthesis | Unknown | 0.5g/q day | 35 | 11 | None | 1.1 | 15.3 | Trace | AKI onset 14 days after Van stop | 1 | Dialysis for 11 days after AKI, lost to follow-up | AKI onset 14 days after Van stop for skin rash. Renal Bx no VNT changes |
| 29 | M | 72 | HTN, COPD, Athero, gout | Hemodynamic, drugs, myoglobinemia | Cellulitis, elbow | SA | 1 g/ 12hrs | 28 | 13 | Clindamycin, sulfamethoxazole/ trimethoprim. | 0.9 | 11.5 | 1+ | AKI onset 21 days after Van stop | 6 | Cret 3 at 31 days, lost to follow-up | AKI onset 21 days after Van stop for skin rash. Renal Bx no VNT changes |
| 30 | M | 63 | DM, HTN, HL, CHF, morbid obesity, pulmonary hypertension | Drugs | Osteomyelitis in femur | Serratia | 1g/q12 hrs | 50 | 23.6 | Cefepime, Piperacillin/ tazobactam | 1.2 | 6.3 | 2+ | 19 | 8 | Back to baseline at 17 days when still on Van | AKI onset 26 days after Van start; recovery even still on Van. Three other episodes of Van treatment not associated with AKI. Renal Bx no VNT changes |

(*Continued*)

**Table 1.** (Continued)

| Case # | Gender | Age (yrs.) | Background Diseases | Possible Causes of AKI, other than VNT and Infection | Primary infection | Culture report | Vancomycin Dose | Duration (days) | Trough Blood Levels of VAN (Microgram/dL) | Other Antibiotics | Creatinine (mg/dL) Baseline | Creatinine (mg/dL) Peak or at Bx | Proteinuria at Bx | Interval between Van Start and AKI (days) | Interval between AKI and Bx (days) | Interval between stopping Van and baseline Serum Creatinine or Last Follow-up Serum Creatinine (days) | Diagnostic Considerations |
|---|---|---|---|---|---|---|---|---|---|---|---|---|---|---|---|---|---|
| 31 | M | 55 | DM, HTN, morbid obesity | Drugs | Scrotal abscess | SA | 2g/q 12 hrs. first day, then 1.5 g/q 12 hrs. | 5 | 15.7 | Amoxicillin and clavulanate, metronidazole | 0.5 | 10.9 | P/C 0.25 | AKI onset 9 days after Van stop | 1 | Back to baseline in 36 days | AKI onset 9 days after Van stop for concern of nephrotoxicity. Another episode of Van treatment not associated with AKI. Renal Bx no VNT changes |
| 32 | F | 51 | DM, HTN, morbid obesity | Drugs, radio-contrast | Foot, septicemia | Streptococcus | 1.75 g/q 12 hrs | 5 | not done | Daptomycin | 0.8 | 3.6 | P/C 0.15 | 5 | 8 | Back to baseline in 60 days | Slow recovery of renal function 60 days). Three other episodes of Van treatment not associated with AKI. Renal Bx no VNT changes |
| 33 | F | 69 | DM, HTN, CHF, stroke | Drugs, hemodynamic, hypothyroism | Urinary tract, septicemia | Negative | 1 g/q 12 hrs. | 10 | 27.9 | Cefepime, meropenem, micafungin | 2.3 | 3.8 | 2+ | 6 | 12 | No recovery, ESRD diagnosed at 21 days | Progression to ESRD after Van stop Renal Bx with immune complex MPGN, |
| 34 | M | 75 | HL, CHF | Drugs, CHF, hemodynamic, myoglobinemia, TMA | Pneumonia | Negative | 1.5 g/q 12 hrs. | 11 | Not done | Cefepime | 1.6 | 3.5 | 1+ | 2 | 12 | No recovery, death due to infection at 41 days | No recovery of renal function after Van stop. Renal Bx with TMA |
| 35 | M | 73 | HTN, HL, sleep apnea, CHF, multiple myeloma | Myeloma cast nephropathy, CHF | Knee prosthesis | MRSA | 1.5 g/day | 5 | Not done | Ondansetron, cephazolin | 0.9 | 6.7 | 2+ | 65 | 2 | Back to baseline in 120 days | AKI onset 65 days after Van start, Recovery 120 days after Van stop. Renal Bx myeloma cast nephropathy |
| 36 | F | 44 | Systemic lupus | TMA | Gluteal abscess | Negative | 1.25 g/12 hrs. | 2 | 27 | None | 1.1 | 3.2 | 2+ | 2 | 6 | Back to baseline in 82 days | AKI persistent after Van stop. Renal Bx with TMA and mesangial lupus nephritis |

NOTES: AKI = Acute kidney injury; ATN = Acute tubular necrosis; Athero = Atherosclerosis; CHF = Congestive heart failure; COPD = Chronic obstructive pulmonary disease; DN = Diabetic nephropathy; IgAGN = IgA nephropathy; MPGN = Membranoproliferative glomerulonephritis; TMA = Thrombotic microangiopathy; IFTA = Interstitial fibrosis and tubular atrophy; DM– Diabetes mellitus; ESRD–End stage renal disease; HL = Hyperlipidemia; HTN = Hypertension MRSA = Methicillin resistant staphylococcus aureus; NA = Not applicable; P/C = Urine protein/ creatinine ratio; SA = Staphylococcus aureus; Van = Vancomycin; VNT = Vancomycin nephrotoxicity

* = Serum creat back to baseline 10 days after discontinuation of van, renal biopsy done 46 days later for heavy proteinuria with normal serum creatinine;

** = The serum creatinine decreased rapidly leading to discharge even when serum creatine was s not back to the baseline/normal levels. Serum creatinine was normal during subsequent hospitalization for unrelated conditions. ** = Serum creat back to baseline 10 days after discontinuation of vancomycin, renal biopsy done 46 days later for heavy proteinuria with normal serum creatinine.

nephrotoxic mediations, severe infection, radio-contrast exposure, hydro-electrolytic derangement, and vancomycin regimen including blood levels. Renal function at baseline, time of most severe AKI, and last follow-up were recorded.

## Results

### Clinical findings of VNT patients (Table 1)

This group included 25 patients. The female(F)/male (M) ratio was 11/14, with an age range of 23–78 years. Diseases with potential nephrotoxic effect were noted in virtually every patient, most frequent among which were hypertension, diabetes, hyperlipidemia, atherosclerosis, and obesity. In each patient, at time of vancomycin treatment, there were several conditions that could cause AKI. Aside from severe infection, these factors included multiple potentially nephrotoxic drugs, radiocontrast administration, hemodynamic stress including dehydration, bleeding, and hypovolemia. The infection was severe and involved a large variety of organs and tissue, most frequent among which was skin and deep soft tissue. Tissue culture positive Staphylococcus was noted in 10 cases with 7 showing methicillin-resistance, other bacteria (7 cases), or negative (9 cases). Baseline serum creatinine was normal in 23 patients for whom standard vancomycin dose was given for 2–69 days, with highest trough blood levels of 15–86 microgram/dL. Standard dose of vancomycin was also given to two other patients (Cases 4 and 21) with unknown baseline serum creatinine but presented with elevated creatinine (4.4 and 2,2mg/dL) at emergency admission for severe infection, leading to a high trough level of 18 microgram/dL in one of them. One patient received vancomycin alone, whereas the rest of patients received other antibiotics and most frequent among them was piperacillin/tazobactam (14 cases). All patients developed severe AKI with peak serum creatine 1.92–11 mg/dL (mean 5.77 mg/dl), shortly after starting vancomycin with a range of 2–69 days. This interval was within 4 days in 16 patients but were longer in few (30 and 69 days in cases 1 and 13 respectively) (Table 1). Proteinuria was negative or trace in most cases, reaching up to a protein/creatinine ratio of 0.7 in only one patient. Discontinuation of vancomycin was associated with rather rapid improvement of renal function with serum creatinine back to the baseline in perhaps all cases. This interval, documented in 17 cases ranges from 7–97 days. (median 14 days). In 8 cases, (Table 1) patients were discharged when serum creatine was still elevated. However, in each of these patients the rate of deceasing serum creatine was also quite rapid (up to 63% in one day) and the serum creatinine was indeed normal during subsequent hospitalization for other conditions (Table 1). In three patients, one or more episodes of vancomycin treatment before or after the AKI-associated episode were noted and they were not associated with alteration of renal function.

The indications for renal biopsy were a. persistent and even progressive elevated serum creatinine after discontinuation of vancomycin (14 cases), b. decision on the course of vancomycin treatment in face of severe life-threatening infection requiring continuous vancomycin and possible VNT (4 cases), or c. determination the exact causes of AKI (7 cases).

### Clinical findings in NO-VNT patients (Table 1)

This group included 11 patients. The final diagnosis of this group was supported by a composite criterion involving clinical, laboratory and renal biopsy findings, detailed for each of these patients in Table 1 (Cases 26–36). These reasons included: 1) AKI long *after* discontinuation of vancomycin for skin rash or concern for nephrotoxicity, with a normal serum creatinine at time of this discontinuation; 2) treatment with oral vancomycin only; 3) AKI developing before starting vancomycin; 4) very slow recovery of renal function after discontinuation of vancomycin; 5) recovery of renal function when still on vancomycin; and 6) several other

episodes of vancomycin treatment not associated with AKI. These findings were not recognized in the VNT group. Furthermore, as detailed in Table 1, more than one reason was noted from the composite criteria for each of the NO-VNT patients. The renal biopsy findings also help to support the diagnosis of NO-VNT. Although renal biopsy changes due to vancomycin nephrotoxicity have not been well defined in literature, over the course of study of the current 26 patients with typical vancomycin nephrotoxicity, a characteristic and quite uniform pathological changes were identified (see below). These findings were not seen in any of the 11 cases of NO-VNT. The renal biopsies further corroborated the diagnosis of NO-VNT, since they revealed causes of acute kidney injury distinct from vancomycin toxicity including TMA and myeloma cast nephropathy (Cases 34, 35, 36).

The clinical prolife of the NO-VNT patients (Table 1) was in general quite similar to that of the VNT group, including demographic distribution, clinical setting, background renal status, and treatment, including vancomycin strategy. However, some notable differences were noted. Aside from the aberrant clinical course correlating vancomycin treatment and renal problem in NO-VNT patients noted above, the final renal outcome of these patients in NO-VNT group was much less favorable than the VNT patients. At least four patients developed progressive renal failure or end-stage renal disease (Cases 26, 27, 33, and 34), whereas the other took a very long duration for return of baseline renal function (36–120 days).

## Renal biopsy findings in VNT patients (Table 2)

LM: Interstitial inflammation was identified in each biopsy. The inflammation was noted in cortical tissue in all cases, but also often seen in medulla, albeit with less severity. It was widespread in most biopsy but was also focal (score 3, 2, and 1 in 9, 12, and 4 biopsies respectively). The interstitial inflammation appeared as scattered predominantly lymphoid cells in each biopsy, with few plasma cells, eosinophils, and rare neutrophils in most biopsies. Against this diffuse background inflammation, many biopsies displayed multifocal inflammatory cell clusters, which were composed predominantly of small lymphoid cells, and plasma cells in variable numbers. Eosinophils were almost always identified within these clusters, often abundant in several of these clusters. Neutrophils, often rare, were noted in a few clusters. These clusters were seen in both cortical and medullary tissue. Most of these clusters were independent, but some of them were closely associated with the tubular segments with vancomycin casts. A rare cluster displayed a perivenous location. In a few biopsies, lymphoid cell nodules with or without reactive germinal centers were seen. Granulomatous inflammation was noted in 6 biopsies. The granulomas were closely associated with damaged tubules, non-necrotic, and negative for acid fast bacilli or fungi by histochemical stains. Tubulitis or peritubular capillaritis was not seen in most cases and when present was inconspicuous. There was marked and diffuse interstitial edema, with no or mild interstitial fibrosis in most cases (Fibrosis score 0, 1–2, and 3 in 15, 9, and 1 biopsy respectively) (Fig 1).

Acute tubular cell injury was seen in each biopsy and was severe in most (ATN score 3, 2, and 1, in 18, 5, and 2 biopsies respectively. These changes preferred proximal tubules and displayed usual features common to ATN including tubular dilatation, flattened tubular cells, loss of differentiation, cytoplasmic vacuolization, and cellular disruption. Of note, these changes were not often associated with interstitial inflammation. Acute tubular injury was also noted less frequently in other tubular segments, especially in distal nephron segments. This type of acute tubular injury was characterized by accumulation of necrotic cells in tubular lumens, often in association with uromodulin casts or vancomycin casts or both, peritubular or luminal accumulation of inflammatory cells including lymphocytes, plasma cells, eosinophils, and neutrophils, or faint granulomatous inflammation. Immunohistochemical stain for Mib-1 showed

**Table 2. Summary of histopathologic features in renal biopsy from patients with and without vancomycin nephrotoxicity.**

**Table: Renal Biopsy Changes**

| Case # | Glomerular Changes | Acute vs Chronic Injury | Interstitial Inflammation (0–3) [a] | Cell Types | ATN [a] (0–3) | IFTA [a] | Myoglobin Cast | Uromodulin Cast | Vancomycin Cast (0–3) [a] | Vascular Changes [a] |
|---|---|---|---|---|---|---|---|---|---|---|
| **VANCOMYCIN NEPHROTOXICITY** | | | | | | | | | | |
| 1 | No changes | A | 2 | LP > N | 3 | 0 | No | Yes | 1 | 2 |
| 2 | No changes | A >>>C | 2 | L > P, N, E | 3 | 1 | Yes | Yes | 2 | 1 |
| 3 | No changes | A | 1 | L > N | 3 | 0 | No | Yes | 3 | 0 |
| 4 | [a] DN I | A>>>C | 1 | LP > E, N | 3 | 3 | Yes | Yes | 1 | 3 |
| 5 | No changes | A | 3 | LP > N; Gr | 3 | 0 | No | Yes | 3 | 2 |
| 6 | No changes | A | 2 | LP > E | 3 | 0 | No | Yes | 3 | 0 |
| 7 | DN IIb | A+C | 1 | LP > E | 3 | 1 | No | Yes | 3 | 1 |
| 8 | No changes | A | 2 | LP = E; Gr | 1 | 0 | No | Yes | 3 | 1 |
| 9 | DN IIa | A+C | 2 | LP, > E, Gr | 2 | 2 | No | Yes | 2 | 3 |
| 10 | DN III | A+C | 2 | LP > E, N | 2 | 2 | Yes | Yes | 3 | 3 |
| 11 | DN I | C | 2 | LP | 1 | 2 | No | Yes | 0** | 2 |
| 12 | DN IIa | A | 3 | LP > E, N | 3 | 2 | No | Yes | 3. | 3 |
| 13 | No changes | A | 2 | L | 3 | 0 | Yes | Yes | 3 | 0 |
| 14 | No changes | A>>>>C | 3 | LP, many E | 3 | 2 | No | Yes | 3 | 2 |
| 15 | DN I | A>>>C | 3 | LP > many E | 3 | 1 | No | Yes | 3 | 2 |
| 16 | DN I IgA GN | A>>>C | 3 | L, many E | 3 | 2 | No | Yes | 3 | 3 |
| 17 | IgA GN | A>>>C | 3 | LP > E, N | 3 | 1 | No | No | 2 | 0 |
| 18 | No Changes | A | 1 | L > E, N | 3 | 0 | No | No | 3 | 1 |
| 19 | IgA GN | A >>>>C | 2 | LP > E | 3 | 0 | No | Yes | 3 | 2 |
| 20 | No changes | A only | 3 | LP >E | 2 | 0 | No | No | 3 | 2 |
| 21 | No changes | A | 3 | LP; Gr | 3 | 0 | Yes | Yes | 3 | 2 |
| 22 | No changes | A >>>>C | 2 | LP > E | 3 | 0 | No | Yes | 3 | 0 |
| 23 | No changes | A | 3 | LP, many E: Gr | 2 | 0 | No | Yes | 1 | 0 |
| 24 | DN II a | A | 2 | L > P, E, N | 2 | 0 | No | Yes | 1 | 1 |
| 25 | DN III | A >>>>C | 2 | L > P, E; Gr | 3 | 0 | No | Yes | 3 | 1 |
| **NO VANCOMYCIN NEPHROTOXICITY** | | | | | | | | | | |
| 26 | DN III/IV | C>>>>A | 0 | L, few P | 1 | 3 | No | Yes | 0 | 3 |
| 27 | DN IIb | A+C | 0 | LP | 1 | 3 | No | Yes | 0 | 3 |
| 28 | No changes | A | 3 | LP, many E, few N | 0 | 0 | N0 | No | 0 | 2 |
| 29 | No changes | A | 3 | LP + many E | 1 | 0 | No | Yes | 0 | 3 |
| 30 | DN I | A | 0 | LP, rare E, rare N | 2 | 1 | No | Yes | 1 | 1 |
| 31 | DN IIa | A | 1 | LP > E, N | 3 | 2 | No | Yes | 3 | 2 |
| 32 | DN II b | A+C | 0 | NA | 3 | 2 | No | Yes | 3 | 2 |
| 33 | MPGN | C>>>>A | 0 | Rare L | 2 | 3 | No | Yes | 0 | 3 |
| 34 | TMA | C>A | 0 | Rare L | 2 | 3 | Yes | No | 1 | 2 |
| 35 | No changes | A | 0 | Rare L | 3 | 0 | Yes | Yes | 3 | 2 |
| 36 | TMA | A | 0 | Rare L | 2 | 0 | Yes | Yes | 2 | 0 |

Footnotes: A–denotes acute histopathological changes, primarily indicating severity of acute tubular necrosis and density of interstitial inflammation in tubule-interstitial nephritis; C–denotes chronic histopathological changes, primarily indicating scarring of the renal cortical compartments, viz., glomerulosclerosis, tubular atrophy, interstitial fibrosis and arteriosclerosis; A>>>>C–denotes significantly more acute changes in a setting of focal chronicity; C>>>>>A–denotes significantly more chronicity changes in renal cortex in a setting of focal acute changes; A + C–shows approximately equal admixture of both acute and chronic changes in renal cortex.

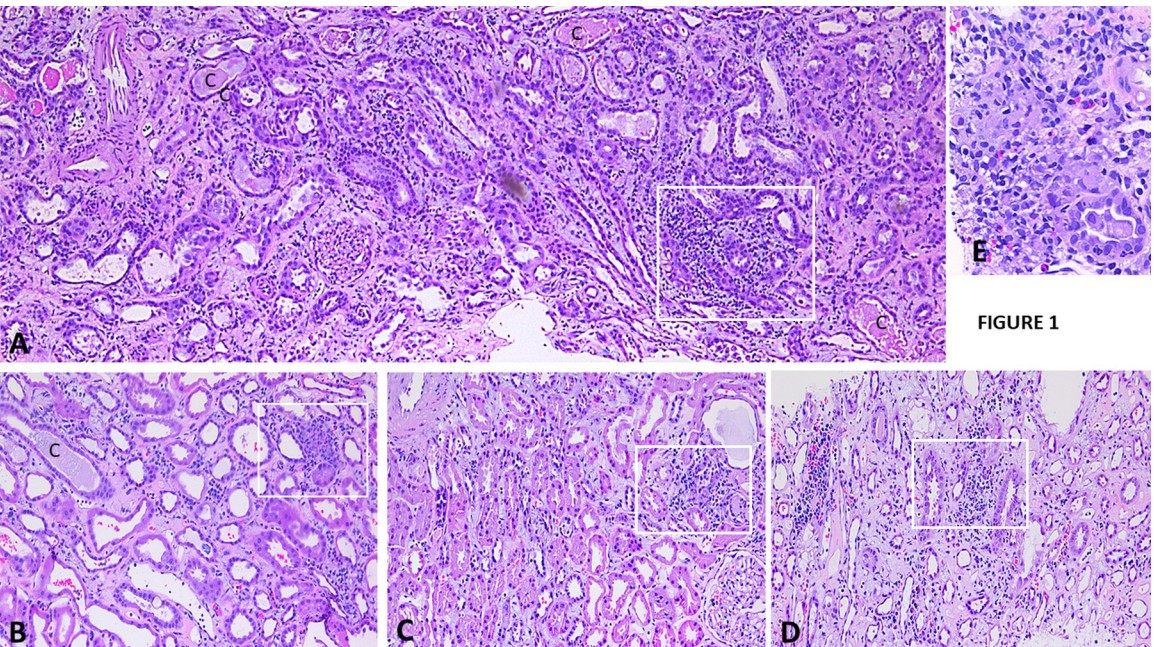

**Fig 1. Highlights the spectrum of tubule-interstitial inflammation in the subset of patients with VNT, highlighting the different types of inflammatory response, in the vicinity of distal tubules with vancomycin casts in cortex as well as in region away from tubules with casts.** Interstitial inflammation. A: There is diffuse moderate interstitial inflammation composed predominantly of lymphoplasmacytic cells forming focal clusters (square), acute tubular injury, and multiple vancomycin tubular casts (C) (Hematoxylin & eosin, x 40). B: Patchy cortical mild interstitial lymphoplasmacytic infiltration with few eosinophils. A cluster of inflammatory cells with several eosinophils is present (square). A vancomycin tubular cast is present perhaps in distal convoluted tubule (C). The proximal tubules show changes consistent with acute tubular necrosis in the absence of tubular casts. There is interstitial edema, without significant tubular atrophy or interstitial fibrosis (Hematoxylin & eosin, x 100). C: Another biopsy with mild interstitial inflammation forming cluster (square). There is mild acute proximal tubular cell injury, without significant tubular atrophy or interstitial fibrosis. A normal glomerulus is present (right lower corner) (Hematoxylin & eosin, x 100). D: Medullary tissue in the same biopsy as in C with the same changes. E: Granulomatous inflammation is noted near to a tubule with possible vancomycin cast (Hematoxylin & eosin, x 100).

several positive tubular cell nuclei in the areas of acute tubular cell injury. Immunohistochemical stain for vancomycin showed vancomycin not only in tubular casts but also in tubular cell cytoplasm, most pronounced in the proximal tubules with acute injury (Fig 2).

Tubular casts of different types were noted in every biopsy. There was a distinctive type of cast composed predominantly of vancomycin confirmed by immunohistochemistry (vancomycin cast). These casts appeared on hematoxylin & eosin stain as microparticles of variable sizes, ranging from finely granular to larger globules with peripheral condensation and a paler center, forming interconnected aggregates of variable sizes. These casts often occupied a portion of tubular lumen, but could be uniform and filled the entire lumen, creating a finely granular appearance. They were faintly PAS positive, with a variegated appearance on silver and trichrome stains. These casts were frequently associated with necrotic (probably tubular) cells or less so with intact or degenerated inflammatory cells including neutrophils. The cells of the tubular profiles that housed these casts often displayed acute injury, as described above. These casts were seen predominantly in the cortical or corticomedullary junction area, and much less frequent in medullary tissue. These casts were localized mostly to the distal nephron segment, especially the thick Henle loops and the distal convoluted tubules, revealed by immunohistochemical stain for tubular segment-specific markers (Figs 3 and 4). These casts were seen in 24/25 biopsies and often numerous (vancomycin cast score 3, 2, and 1 in 17, 3, and 3 biopsies respectively). In one case (Case 11), vancomycin cast was not seen; perhaps reflecting delayed

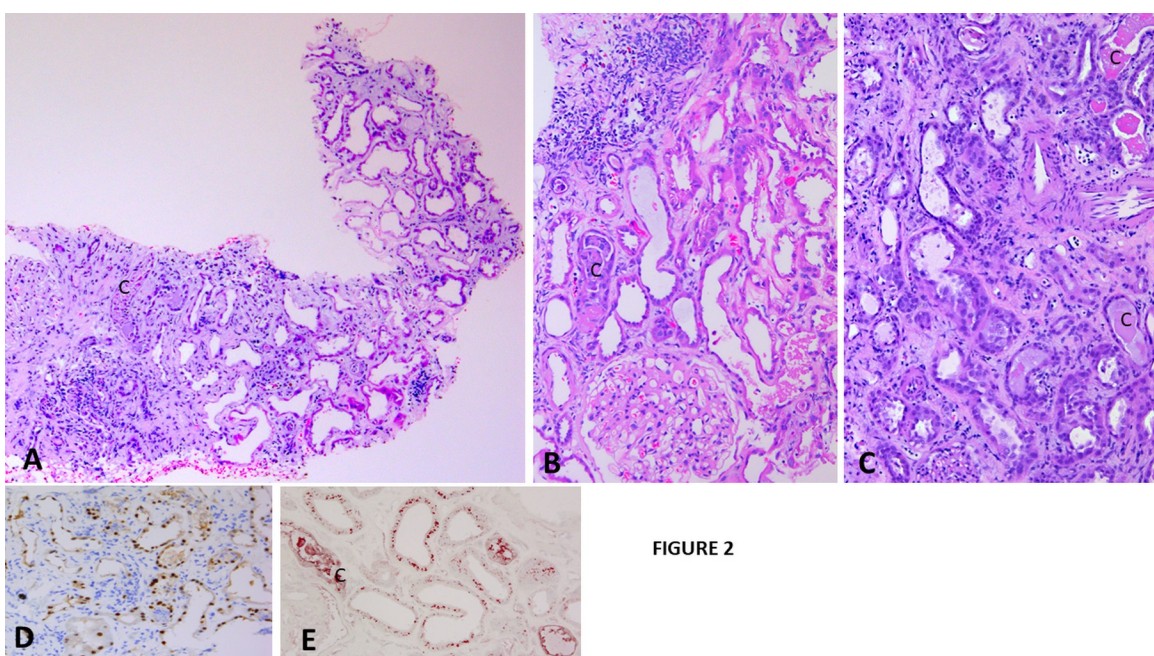

FIGURE 2

**Fig 2. Highlights the tubular necrosis in the subset of patients with VNT.** Acute tubular necrosis. A: There are changes of acute tubular necrosis involving mostly proximal tubules (upper field). These are not associated with significant interstitial inflammation or tubular casts. Interstitial inflammation is also focally noted together with vancomycin tubular casts (c) (lower field) (hematoxylin& eosin, x 40). B: In another biopsy, there are pronounced changes consistent with acute tubular necrosis, without associated tubular atrophy, or interstitial inflammation. A vancomycin cast associated with individual tubular cell necrosis is noted perhaps in a distal nephron segment (hematoxylin & eosin, x 100). C: Another biopsy showing acute tubular necrosis, associated with mild interstitial inflammation. Several vancomycin casts are noted (c) (Hematoxylin& eosin, x 100). D = Several tubular cell nuclei in the area of acute tubular necrosis show nuclear expression of Mib-1 (Immunostain, x 100). E = Vancomycin is detected in proximal tubular cell cytoplasm in area of acute tubular injury, but there is no vancomycin cast in them. In contrast, a vancomycin cast (C) is noted in the lumen of a tubular segment perhaps not proximal tubule (Immunostain x 100).

biopsy done to evaluate significant proteinuria, 46 days after discontinuation of vancomycin. Other types of tubular casts were also present. Aside from the hyaline casts, uromodulin casts were noted in 22/25 biopsies and myoglobin casts in 5/25 biopsies.

Glomerular changes, almost always minor, were noted including no significant alteration (13 biopsies); diabetic glomerular changes (WHO Class I, IIa, IIb, and III in 4, 3, 1 and 2 biopsies respectively); and incidental IgA nephropathy detected only by IF study in three biopsies.

The blood vessels showed mild arteriosclerosis or no significant changes.

IF: The IF study showed no immunoglobulins or complement components in the tubulointerstitial compartment of any of the biopsies. Findings in support of the light microscopic glomerular or vascular changes were found.

EM: The EM study revealed distinctive and characteristic feature of vancomycin tubular casts. Several forms of these cast were identified, perhaps reflecting stages of formation. The fully developed casts appeared as individual or aggregated spherules with well-defined smooth contour and a circumferential and orderly concentric lamellar formation involving the outer portion of the cast or its entire area. The EM study corroborated several light microscopic findings including frequent co-deposition with uromodulin and or necrotic cell debris, preferential location in distal nephron segment and acute tubular cell injury involving the tubular segment housing the casts and proximal tubules even in the absence of cast in these tubules. Electron dense deposits were not identified in the tubulointerstitial compartment in any biopsy. EM findings in support of the LM glomerular changes were noted.

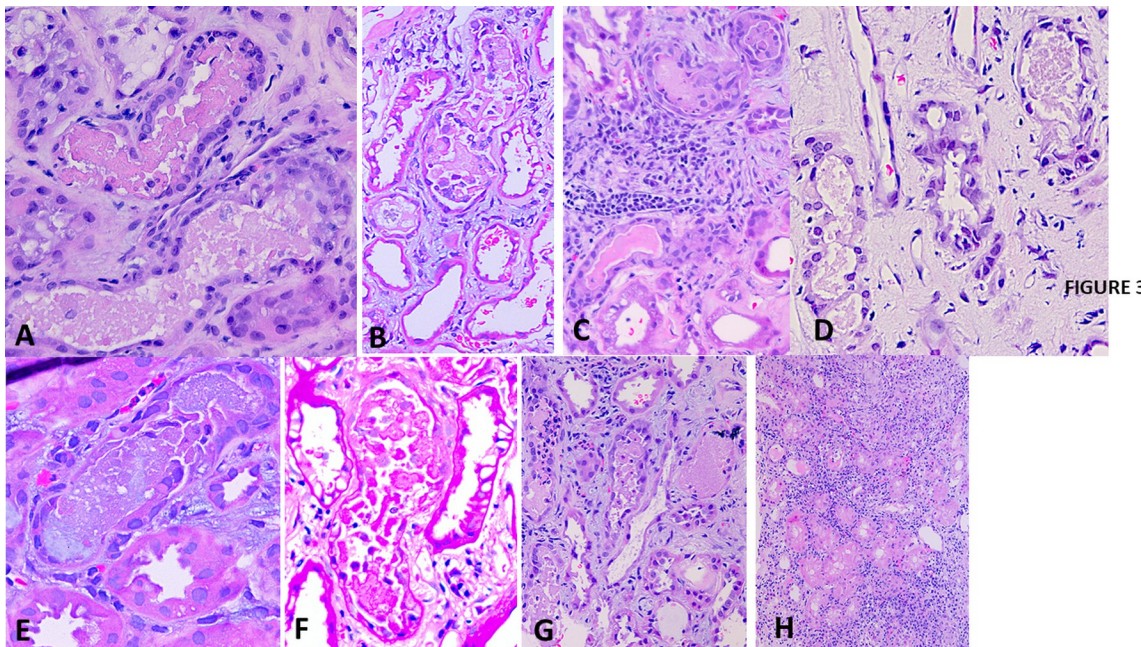

**Fig 3. Vancomycin tubular casts as highlighted in hematoxylin & eosin and PAS stains.** A: Vancomycin tubular casts with a distinctive morphologic spectrum are noted (Hematoxylin & eosin x 400). B: A vancomycin tubular cast associated with necrotic tubular cells. The adjacent proximal tubules show changes of acute tubular cell injury, without associated vancomycin cast (Hematoxylin & eosin x 400). C = A vancomycin cast in the upper field is associated with necrotic cells and rare neutrophils, adjacent to a cluster of interstitial inflammatory cells. Another tubular cast in the lower field, probably also vancomycin casts does not show secondary changes (Hematoxylin & eosin x 400). D = Vancomycin tubular casts in medullary tubules perhaps thick Henle loops (Hematoxylin & eosin x 400). E = A vancomycin cast with coprecipitation of uromodulin appearing as homogeneous bluish material. Adjacent proximal tubules are intact (Hematoxylin & eosin x 400). F = A vancomycin tubular cast under PAS stain highlighting the lamellar formation and a faint PAS positive staining for the cast, Adjacent proximal tubules show acute injury (PAS stain, x400). G- Vancomycin tubular cast in different forms in the same field: multi-globular (lower left), diffuse (upper right), associated with inflammatory and necrotic tubular cells (upper left) (Hematoxylin & eosin x 400). H—Acute tubulointerstitial nephritis in a patient with renal biopsy done 21 days after discontinuation of vancomycin for skin rash when the serum creatinine was normal (Control slide for vancomycin toxicity). Severe and diffuse interstitial inflammatory cell infiltrate, with prominent eosinophils, but without acute tubular necrosis or vancomycin cast (Hematoxylin & eosin, x 100).

### Renal biopsy finding in NO-VNT patients (Table2)

Among the kidney biopsies in the 11 cases with the final diagnosis of NO-VNT, two (Cases 28 and 29) showed severe acute TIN, including diffuse and interstitial back-to-back inflammatory cell infiltrates including many eosinophils, severe tubulitis, mild or no ATN, and absence of inflammatory cell clusters or vancomycin casts or tubular accumulation of vancomycin even by immunohistochemical study (Fig 4D); these changes were quite different from the characteristic vancomycin-related TIN described above. In these two patients, vancomycin was stopped due to skin rash. At that time their renal function was at baseline, but AKI then developed 14 and 21 days later for which kidney biopsy was repeated. Four biopsies showed specific disease process unrelated to vancomycin toxicity including thrombotic microangiopathy (2), myeloma cast nephropathy (1), and immune-mediated MPGN (1). The five remaining biopsies showed ATN of variable severity, with no or minimal interstitial inflammation, but with moderate or marked interstitial fibrosis in all of them. The acute tubular injury in these biopsies did not show several of the distinctive features of its vancomycin-associated counterpart described above. Diabetic glomerular changes were noted in each of these biopsies. Vancomycin casts were seen in 6/11 biopsies and casts were numerous in 3 biopsies (vancomycin cast score 3, 2, and 1 in 3, 1, and two biopsies respectively). There were also uromodulin casts (9/11

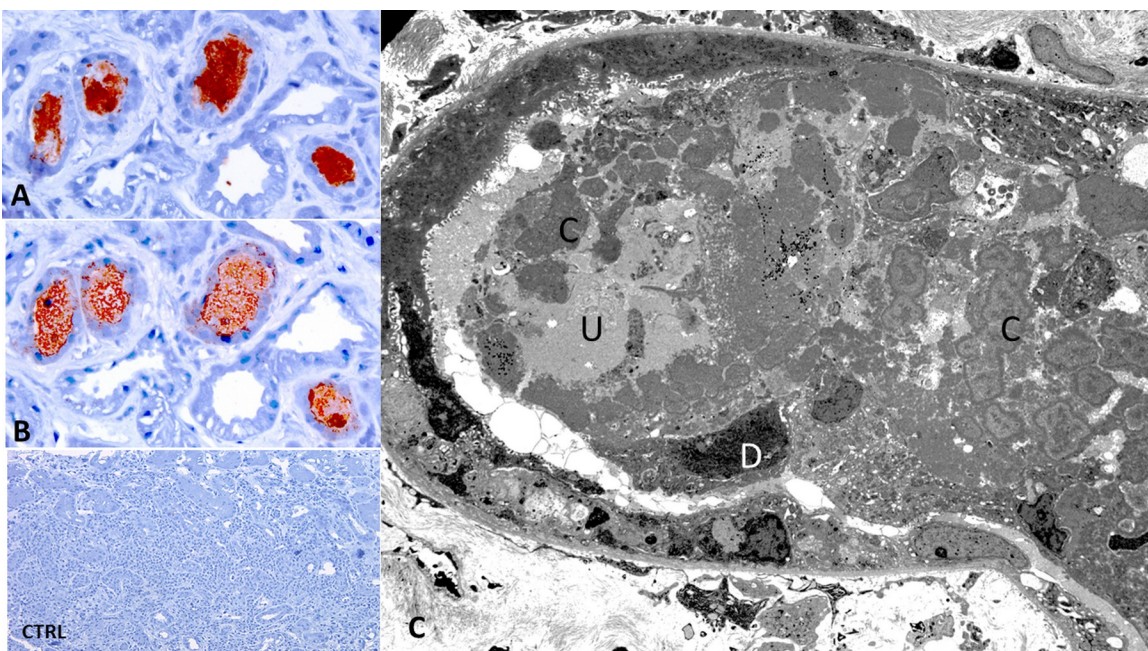

**Fig 4.** A: Vancomycin tubular casts detected by immunohistochemical staining for vancomycin (Immunostain x 400). B = Uromodulin is also detected in the same casts by immunohistochemical staining (Immunostain x 400). C = Electron microscopy showing the characteristic lamellar appearance of vancomycin casts (C). There is also coprecipitation with uromodulin (U) and damaged/necrotic tubular cells (x 8000). D. Immunostain for vancomycin is negative for a control biopsy (Case 29) (Immunostain x 100).

biopsies) and myoglobin casts (2/11 biopsies). The IF and EM studies supported the above diagnostic changes.

## Discussion

Vancomycin is a highly hydrophilic glycopeptide antibiotic that is increasingly used to treat severe infection especially for the methicillin-resistant Staphylococcus aureus. In fact, vancomycin is the most frequently prescribed antibiotic in the hospital setting, for up to 35% of hospitalized patients with infection. Since it is not absorbed by oral route, vancomycin is administered intravenously with a recommended dosage aiming to reach a therapeutic trough blood level of 5–15 microgram/dL. Vancomycin is metabolized in liver (<5%), but mostly follows the glomerular filtrate and is secreted unchanged in urine, along with minimal reabsorption and metabolism by proximal tubular cells. Kidney injury associated with vancomycin treatment, i.e., VNT, is well recognized as the most significant treatment side effect. Although VNT is quite frequent, being reported in 5–35% of treated patients, its clinical, morphologic, and pathogenetic profiles is in fact still poorly understood, due to subject matter limitation, including the inclusion of cases with equivocal diagnosis, multiple confounding conditions, or lack of biopsy confirmation [8, 9].

The current study focuses on the cases in which the diagnosis of VNT is reasonably certain. Although the case number is rather small, they help establish VNT as a characteristic clinico-pathologic entity. The VNT patients are often elderly individuals, almost always affected with several systemic diseases well recognized to be nephrotoxic, including diabetes, hypertension, and atherosclerosis. In spite of this background, the renal function seems to be well preserved in them as indicated by a normal baseline serum creatinine. Thus, when severe infection supervenes, vancomycin, as the antibiotic of choice, is given in standard dose. This treatment

is followed by severe AKI, with a marked and quick elevation of serum creatinine from a normal base line, which develops quickly, sometime as early as one day after treatment. This may happen even before vancomycin blood level becomes available as a guidance for management including dose adjustment. There is no or only insignificant proteinuria in almost all cases. This characteristic precipitous course, previously suggested by Velez et al, is confirmed in this study [10]. Discontinuation of vancomycin, upon judicious clinical decision and/or the characteristic renal biopsy findings, is associated with rather quick recovery of renal function to the baseline, without progression to chronic kidney injury. Some of the renal changes seen in renal biopsies may account for this course. In almost all biopsies there are both acute TIN and an independent ATN of nephrotoxic type, without significant glomerular or vascular changes, perhaps by themselves adequately accounting for the accelerated ATN. Furthermore, although interstitial edema is ubiquitous, established interstitial fibrosis is not seen or quite mild, serving as a foundation for full recovery.

Against this prototype, other pertinent considerations surface. The vancomycin pharmacodynamics is noteworthy in the context of VNT. The generally accepted therapeutic trough level of vancomycin is 10–20 microgram/dL. The half-life of vancomycin is 4–6 hours in those with normal renal function but can be as long as 7 days in association with renal failure. Since the serum creatinine of all patients in this study was normal or near normal, a standard therapeutic dose was started. Yet, in most of these patients at some time during the treatment course, the vancomycin trough level reached nephrotoxic level (mean 28.29 microgram/dl). Furthermore, this trough level remained quite high in many patients long after discontinuation of vancomycin (median duration 15 days). The cause of this aberrant pharmacodynamics is perhaps multifactorial, but probably also reflecting renal disease-related poor clearance of vancomycin [11–13].

Vancomycin was used alone in only one patient, but in combination with other antibiotics in the rest, most frequent among which was piperacillin/tazobactam (14 cases), raising the concern on their roles in the pathogenesis and morphology of VNT [14]. Previous studies suggested that piperacillin/tazobactam enhances VNT, mostly through a proximal tubular nephrotoxic effect synergistic with that of vancomycin [15, 16]. This finding, however, is not supported by the more recent critical analysis. Piperacillin/tazobactam alone has not been documented to be nephrotoxic but may cause spurious elevated serum creatinine by reducing renal excretion of creatinine through the organic anion transport system. These contentions may be supported by our study in which the clinical course and the renal biopsy findings of those with and without piperacillin/tazobactam are quite similar [17–21].

The role of renal biopsy in the management of VNT is unsettled. In most large-scale studies of VNT, it was reported that the vancomycin-associated AKI was managed by discontinuation of the drug, leading to clinical improvement, thus confirming the diagnosis of VNT without resorting to renal biopsy. This approach, perhaps justified in most cases, especially in those with milder AKI, may leave a level of diagnostic uncertainty in the subjects, and at the same time, denies a pathway to study the pathogenesis of VNT through biopsy examination. Renal biopsy was done for each patient in our study, mostly in response to a need to know the exact cause of the often severe and progressive AKI even after discontinuation of vancomycin. In several other cases, there is a critical decision to make on whether vancomycin should be continued or stopped in the face of both severe life-threatening infection and AKI potentially caused by vancomycin and renal biopsy is essential for this decision. On the other hand, in several cases in our study, the renal function had already normalized or displayed a rapid improvement when the renal biopsy result became available. Thus, in retrospect, the need for renal biopsy in this setting needs a consideration based on individual patient's clinical

condition. An awareness of the quite characteristic profile of VNT, described in our series, may help in this decision.

Assessment of a distinct clinical profile of VNT may be facilitated by a simultaneous study of those who were treated with vancomycin, but in whom AKI was finally deemed not to be due to vancomycin. Our study shows that although the renal outcome is quite different in these two settings, other findings related to demography, background disease, and clinical course were quite similar. These observations underline a need for renal biopsy in this clinical context, as well as the related question on whether the renal biopsy changes are specific for a diagnosis of VNT.

The morphologic changes of kidneys associated with VNT have not been identified adequately. There are about 30 isolated case reports in which ATN or acute TIN, or both have been rather briefly reported. In a few reports, the illustration did show vancomycin cast, but this finding was indeed not mentioned by its name. These reports provide some morphologic insights, but leave unanswered a comprehensive morphologic spectrum and, equally important, whether this spectrum is indeed specific of VNT, that can facilitate both diagnostic and pathogenetic approach [22–24]. Our comprehensive renal biopsy study suggests that there is such a VNT-specific morphologic spectrum. In almost all biopsies in our study, there is concomitant acute TIN of allergic type, ATN of nephrotoxic type, and vancomycin casts, albeit in variable severity for each biopsy (Table 2). Against this background of severe acute tubulointerstitial changes, there is no significant tubular atrophy, interstitial fibrosis, or glomerular changes. This distinctive morphologic profile not only affords an accurate biopsy diagnosis, but also accounts for the clinical course, and provides pathogenetic insight.

The acute TIN in VNT is characterized by interstitial inflammatory cell infiltrate in the absence of immunoglobulins/complement components or electron dense deposits in the tubulointerstitial compartment, as expected for most types of acute TIN. Furthermore, there is eosinophil infiltration displaying a quite distinctive pattern in which aside from the scattered eosinophils as a component of the diffuse interstitial inflammation, eosinophils may aggregate as clusters in conjunction with other inflammatory cells. This pattern of eosinophilic infiltration, to the best of our knowledge, has not been documented in other types of acute TIN, even for the drug-induced type. Although eosinophils are well documented as a distinctive feature of drug-induced acute TIN in general, the magnitude may vary from drug-to-drug, and thus the infiltration can be quite pronounced as for methicillin, but rather inconspicuous in nonsteroidal anti-inflammatory drugs. Furthermore, eosinophil infiltration may be fleeting and thus may not be obvious in late biopsy. Against this background, the constant eosinophil infiltration in the current biopsies not only reflects a distinctive VNT-related change but may also be due to the quite short interval between AKI and renal biopsy (Table 2). There was also granulomatous inflammation. This type of inflammation, most frequently due to drug reaction among the long list of possible etiology, was indeed quite frequent (6/25) in VNT, a finding not necessarily specific, but certainly supporting an allergic drug-induced etiology for this condition.

ATN is always present albeit of variable severity in VNT. Along the recognized border of nephrotoxic vs ischemic type of ATN, the ATN associated with VNT is predominantly of nephrotoxic type, with a characteristic change focusing on proximal tubules, as detailed above. The nephrotoxic nature of the ATN is further supported by the absence of a significant tubulointerstitial inflammation in the affected tubular segments as well as immunohistochemical presence of vancomycin in these tubules. The nephrotoxic ATN may account for the quite rapid onset of AKI, in contrast to the potential delayed onset of AKI due to drug allergy, that usually requires time for the drug-related T cell immunologic response to develop. Subtle acute tubular injury has been documented in ATN of different etiology and are thought to be

due to the associated cytotoxic T-cell infiltration, but a pronounced ATN of nephrotoxic type, to the best of our knowledge, has not been reported, Thus the distinctive pattern of nephrotoxic ATN in the current biopsy implies a specific diagnostic and pathogenic changes for VNT.

Vancomycin tubular casts are an important histopathologic change that are almost always seen in renal biopsies in patients with VNT. However, vancomycin tubular casts are not entirely specific for VNT, since it is also noted in patients in the NO-VNT group. Vancomycin casts were noted in 24/25 biopsies in VNT group, often in abundance (score 3 in 16/25 biopsies, Table 2). In one patient (Case 11), vancomycin casts were not seen in a biopsy done for significant proteinuria at normal serum creatine, 46 days after discontinuation of vancomycin. These casts are composed of tubular accumulation of vancomycin often co-precipitated with uromodulin, as confirmed by immunohistochemistry. It has a distinctive, perhaps almost pathognomonic appearance, but may simulate myoglobin casts. Myoglobin immunohistochemistry may be helpful in the differential diagnosis. This was done in all biopsies and revealed myoglobin casts in five biopsies; however, in each case, only a few casts were myoglobin-positive, but many other adjacent presumably vancomycin casts were indeed negative. This is an observation of considerable diagnostic implication since myoglobin can cause AKI by itself in the contest of polypharmacy, and myoglobin immunohistochemical stain is readily available as a diagnostic test, but immunohistochemical staining for vancomycin is only done in research setting. Vancomycin casts, sometime quite pronounced, were noted in 6/11 biopsies from NO-NNT group of patients, casting a doubt on its role as the main cause of VNT. Vancomycin cast can serve as an evidence of previous vancomycin exposure/treatment, but may not be critical in the development of vancomycin nephrotoxicity.

The idea that the morphologic spectrum described above is helpful in diagnosis of VNT, is also supported by comparative examination of the renal biopsies in the NO-VNT patients. These changes of VNT, in full combination, were indeed not identified in any of these biopsies. In most of them there was ATN of variable severity, which in combination of other changes may account for AKI, but TIN is almost always absent. In contrast to the negligent interstitial fibrosis and tubular atrophy in VNT patients, cortical scarring is frequent and often severe in the NO-VNT biopsies, adding a differential diagnostic hint, and perhaps accounting for the frequent poor renal outcome of the NO-VNT patients. Of note, allergic TIN did develop in two cases (Cases 28 and 29, Table 2), but the TIN in both cases showed a severe and diffuse interstitial inflammation with prominent eosinophilic infiltration certainly implying an allergic cause, but of a morphology quite different from that of VNT. Furthermore, the clinical course of these two patients including the development of AKI long after vancomycin discontinuation, further corroborate the NO-VNT nature of the acute TIN. These two biopsies indeed lend more support to the specific nature of VNT-associated renal pathologic changes. Vancomycin casts, sometime quite pronounced, were noted in 6/11 biopsies, casting a doubt on its role as the main cause of VNT.

The pathogenesis of VNT is still unknown. Few etiologic pathways of VNT have been described. Vancomycin clearance mostly follows the glomerular filtrate and is secreted unchanged in urine, along with minimal reabsorption and metabolism by proximal tubular cells. Nephrotoxicity may involve overload of tubular cell uptake, promoted by many known risk factors for nephrotoxicity, leading to oxidative stress, generation of reactive oxidative radicals, mitochondrial injury, and finally cell death. This pathogenesis pathway has been noted in several observational and interventional models in rodents and cell cultures. The current human study provide substantial proof for this pathogenesis pathway since ATN of nephrotoxic type in conjunction with abundant proximal tubular vancomycin upload is ubiquitous. Implication of distal tubular segment injury was noted in a single early study describing

elevated level of dimethylamine, a distal nephron segment marker, in some patients treated with "aminoglycoside and/or glycopeptide. Injury to distal tubular segments, noted in our study, albeit not pronounced and often in conjunction with vancomycin casts and cast-associated inflammation, lends some support to this pathway, which perhaps is only of a limited pathogenetic role. Drug allergy emerges as an important pathogenetic element since allergic type of acute TIN is almost always noted with constant eosinophilic infiltration and granulomatous inflammation, common to drug-induced TIN, is also quite frequent. This is a significant observation since this type of "idiosyncratic" injury, perhaps reflecting individual genetic background susceptibility, is not amenable to animal model studies. A pathogenetic pathway implicating a concomitant element of both nephrotoxic ATN and allergic type of acute ATN is thus proposed for VNT, a quite unique pathway that deserves further study. Since the advent of vancomycin casts, its pathogenetic role has been scrutinized. Findings from the current study suggests that vancomycin casts may add to the renal injury but may not be the main culprit. The findings suggest that ATN and or TIN initially may induce AKI, including single nephron obstruction. This leads to poor proximal handling of vancomycin, formation of uromodulin casts, increased local concentration of vancomycin, precipitation/ crystallization of vancomycin, and localized necrosis and damage of tubular epithelial cells, feed into a self-perpetuating positive feedback loop of local tubular injury, superimposing on the ATN and/or TIN-induced background AKI. This conclusion is also supported by the observation that vancomycin casts were frequently observed in the biopsy from those who were treated with vancomycin but did not develop VNT [12, 13, 15, 16].

Our study is limited by several considerations, which, due to its design and nature, could not have been effectively addressed. These limitations include a single-center design, limiting the generalization of the results; a selection bias with rather small sample size, and inclusion of only cases with severe disease requiring renal biopsies; criteria for diagnostic assignment remain empirical and subject to interpretation; retrospective nature with limited follow-up duration, as well as cases loss for follow-up.

In summary, VNT has a distinctive morphologic and clinical profile, which should facilitate diagnosis, guide treatment and prognostication, and confer pathogenetic insights.

## Author Contributions

**Conceptualization:** Rajesh Nachiappa Ganesh, Luan D. Truong.

**Data curation:** Rajesh Nachiappa Ganesh, Roberto Barrios.

**Formal analysis:** Rajesh Nachiappa Ganesh, Angelina Edwards, Ziad El Zaatari, Lillian Gaber, Roberto Barrios, Luan D. Truong.

**Investigation:** Rajesh Nachiappa Ganesh, Ziad El Zaatari, Lillian Gaber, Roberto Barrios, Luan D. Truong.

**Methodology:** Rajesh Nachiappa Ganesh, Angelina Edwards, Ziad El Zaatari, Lillian Gaber, Roberto Barrios, Luan D. Truong.

**Supervision:** Luan D. Truong.

**Validation:** Rajesh Nachiappa Ganesh.

**Writing – original draft:** Rajesh Nachiappa Ganesh.

**Writing – review & editing:** Rajesh Nachiappa Ganesh, Roberto Barrios, Luan D. Truong.

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
