## [Decision Letter · Decision Letter 0]

8 Aug 2023

PONE-D-23-17096Vancomycin Nephrotoxicity: A Comprehensive Clinico-pathological StudyPLOS ONE

Dear Dr. Nachiappa Ganesh,

Thank you for submitting your manuscript to PLOS ONE. After careful consideration, we feel that it has merit but does not fully meet PLOS ONE’s publication criteria as it currently stands. Therefore, we invite you to submit a revised version of the manuscript that addresses the points raised during the review process.

Please, address the issues raised by Reviewer 1 and acknowledge in the discussion the  limitation of the study as the small sample of analyses patients and the  single center design, as well as its retrospective nature.

We look forward to receiving your revised manuscript.

Kind regards,

Fabio Sallustio, PhD

Academic Editor

PLOS ONE

Journal Requirements:

"NO"

"No authors have competing interests"

Reviewers' comments:

Reviewer's Responses to Questions

**Comments to the Author**

1. Is the manuscript technically sound, and do the data support the conclusions?

Reviewer #1: Yes

Reviewer #2: Yes

2. Has the statistical analysis been performed appropriately and rigorously? 

Reviewer #1: N/A

Reviewer #2: Yes

3. Have the authors made all data underlying the findings in their manuscript fully available?

Reviewer #1: Yes

Reviewer #2: Yes

4. Is the manuscript presented in an intelligible fashion and written in standard English?

Reviewer #1: Yes

Reviewer #2: Yes

5. Review Comments to the Author

Reviewer #1: Ganesh, et al. conducted a clinicopathologic study of vancomycin nephrotoxicity by evaluating in detail the renal biopsy findings of patients suspected to have vancomycin nephrotoxicity versus those who received vancomycin but were assumed to have nephrotoxicity due to a different etiology. The overall methodology of the study appears to be sound and adds to current literature. There is, however, a need for further clarification of certain aspects of the methods and reconsideration of whether certain NO-VNT cases should be included in the study. Additionally, there are some needed clarifications in the tables and minor typographical errors in the body of the manuscript (primarily sentences ending in commas instead of periods) which need to be addressed.

Major:

1. The exact objective of the study is not clearly stated in the introduction, although it is described in vague terms in the last paragraph of the introduction.

2. When defining the cohort, what was “vancomycin around the time of biopsy” defined as? Was there a specified time period for how long the patients had to have been on vancomycin to qualify for inclusion?

3. The description of the NO-VNT group as a control group (as mentioned in the last sentence of the introduction) is somewhat misleading since it was not only clinical criteria that were used to place patients into separate groups, but rather also the biopsy findings. It is difficult to call patients “controls” when the renal biopsy findings for VNT are still being explored in this study.

4. There needs to be additional description of the criteria to determine whether patients were in the VNT or NO-VNT group. There is some mention of the clinical criteria that were used in the results section, but this should be included in the methods and further detailed. Additionally, information about the process in how patients were categorized is needed (e.g., were cases reviewed by multiple clinicians).

5. The reason for inclusion of certain patients in the NO-VNT group is unclear, such as those who developed nephrotoxicity distantly after discontinuation of vancomycin, since they are not necessarily comparable to the patients receiving renal biopsy in suspected VNT cases. Similarly, a patient who received only PO vancomycin, such as case 27, should not be included either since PO vancomycin has minimal systemic absorption. An example of how this may bias findings is how cases 27, 28, and 29 have no evidence of vancomycin casts. If only such patients were included in the NO-VNT group, a misleading conclusion that vancomycin casts are critical to the pathogenesis of VNT could be reached.

6. It is not clear to me how vancomycin casts can be considered a “specific change of VNT” when there was also a significant proportion of patients in the NO-VNT group who also had vancomycin casts on biopsy (particularly if you exclude cases 27, 28, and 29).

7. There is no mention of the limitations of the study, such as selection bias given only patients with more severe or unclear cases of VNT received biopsies. A discussion of the limitations of the study should be included.

Minor:

1. In the below sentence from the discussion, was “pathogenetic” was meant to be “pathogenic”? It is not clear how the study would show “pathogenetic” changes for VNT.

“Thus the distinctive pattern of nephrotoxic ATN in the current biopsy implies a specific diagnostic and pathogenetic changes for VNT.”

2. In Table 1, it is not clear what “typical clinical course” means. The description of what a typical clinical course is alluded to within the body of the manuscript, when discussing the rapid improvement, but it is not clearly defined at any point. “Typical renal biopsy” is more clearly described in the body of the manuscript given the extensive discussion regarding the biopsy findings, but information in the legend regarding what this means may be helpful.

3. In Table 1, there is only a single vancomycin dose and vancomycin trough level listed per case. The dose of vancomycin may change based on the trough levels and trough levels are expected to vary. Were only the maximum dose and trough level reported?

4. How was baseline creatinine defined?

5. In Table 2, it is not clear what “A>>>C” means. “A+C” presumably means both acute and chronic injury is present, but this is also not entirely clear.

6. The below sentence from the discussion, is phrased in a manner that is difficult to understand.

“Thus, when severe infection supervenes, vancomycin, as the antibiotic of choice, is given in standard dose, upon which severe AKI occurs, reflected by a highly elevated serum creatinine, develops quickly sometime as early as one day after treatment.”

7. There are several sentences that end with commas rather than periods, which should be corrected.

Reviewer #2: The first limitation of the study is the single center design, which could decrease the generalizability of the results and their external validity, as well as its retrospective nature. Moreover, a further limitation of the study is represented by the small sample of patients examined.

6. PLOS authors have the option to publish the peer review history of their article (what does this mean?). If published, this will include your full peer review and any attached files.

Reviewer #1: No

Reviewer #2: No

---

## [Author Response · Author response to Decision Letter 0]

10 Oct 2023

Dear Editor

 We truly appreciate the editorial decision to allow us to improve our study and resubmit to PLOS for possible publication. 

 We are deeply thankful for a meticulous and insightful review offered but the Reviewers. These comments should not only guide us to improve the scientific merit of our current study, but also offer a general instruction on study design for our future scientific endeavors. 

 Please find below the responses to each of the comments raised by the Reviewers and the resultant revision of the manuscript, when appropriate. 

REVIEWER #1

 Ganesh, et al. conducted a clinicopathologic study of vancomycin nephrotoxicity by evaluating in detail the renal biopsy findings of patients suspected to have vancomycin nephrotoxicity versus those who received vancomycin but were assumed to have nephrotoxicity due to a different etiology. The overall methodology of the study appears to be sound and adds to current literature. There is, however, a need for further clarification of certain aspects of the methods and reconsideration of whether certain NO-VNT cases should be included in the study. Additionally, there are some needed clarifications in the tables and minor typographical errors in the body of the manuscript (primarily sentences ending in commas instead of periods) which need to be addressed.

 Response: We are grateful for a favorable impression on the merit of our study, as well as its limitations. We are sure that this favorable impression is a major reason for the editorial decision to allow us to re-submit this manuscript. 

 Revision: None

Serial number Reviewer comment Authors response

1 The exact objective of the study is not clearly stated in the introduction, although it is described in vague terms in the last paragraph of the introduction.

 We wish to add the following sentences at the beginning of the last paragraph of Introduction to clearly delineate the objective of this study “The current study aims to comprehensively evaluate the clinical manifestations and the renal changes of vancomycin nephrotoxicity”

Page 6, Line 14 and 15

2 When defining the cohort, what was “vancomycin around the time of biopsy” defined as? Was there a specified time period for how long the patients had to have been on vancomycin to qualify for inclusion?

 For case selection, we initially reviewed our record of renal biopsies and identify cases with a clinical notation “vancomycin”. We then reviewed the full clinical histories of these patients and noted that in these cases kidney biopsy was done to evaluate the cause of acute kidney injury in a patient who was treated with vancomycin. The time interval between the onset of vancomycin treatment and renal biopsy ranged from 2-69 days, with a median of 6 days from the initial vancomycin administration

Page number 7, lines 7-13 in Materials and Methods 

3 The description of the NO-VNT group as a control group (as mentioned in the last sentence of the introduction) is somewhat misleading since it was not only clinical criteria that were used to place patients into separate groups, but rather also the biopsy findings. It is difficult to call patients “controls” when the renal biopsy findings for VNT are still being explored in this study. The word “control” is indeed not correct in the current context. The words “a control group” is deleted.

Control group is replaced by “comparative group”

Page 4, line 26 

Page 6, line 11 

4 There needs to be additional description of the criteria to determine whether patients were in the VNT or NO-VNT group. There is some mention of the clinical criteria that were used in the results section, but this should be included in the methods and further detailed. Additionally, information about the process in how patients were categorized is needed (e.g., were cases reviewed by multiple clinicians). Vancomycin is frequently used to treat infection. Acute kidney injury may develop in association with this treatment. There is a need to determine whether the acute kidney injury is due to vancomycin requiring cessation of this treatment or it is unrelated to vancomycin so that vancomycin can be continued. This dilemma not infrequently leads to divergent opinions of the treating nephrologist and the infection specialist. The clinical manifestations and the renal biopsy findings, which may be distinctive for vancomycin nephrotoxicity, may be helpful for this essential differential diagnoses. However, the pertinent current literature is very limited. In an effort to address this matter we have attempted to classify the patients into a group with vancomycin nephrotoxicity and those without vancomycin nephrotoxicity for a comprehensive comparative evaluation. The reasons for this patients group assignment are detailed in the Results, discussion and in Table 1. We are ready to admit that, as pointed out by the Reviewer that this distinction may be difficult, especially at the time of actual patient care. However, we also trust that the retrospective nature of this study would significantly facilitate this distinction. At the recommendation of the Reviewer, we now described the inclusion criteria in more detail in the Material and Methods. 

The pertinent description in the Results is now more succinct, focusing more on the actual data in Table 1. 

Revision made - A new paragraph, most of which was previously placed in the Results, is now added to the Materials and Method “Vancomycin is frequently used to treat infection. Acute kidney injury may develop in association with this treatment. There is a need to determine whether the acute kidney injury is due to vancomycin requiring cessation of this treatment or it is unrelated to vancomycin so that vancomycin can be continued. This dilemma not infrequently leads to divergent opinions of the treating nephrologist and infection specialist. The clinical manifestations and the renal biopsy findings, which may be distinctive for vancomycin nephrotoxicity, may be helpful for this essential differential diagnoses. However, the pertinent current literature is very limited. In an effort to address this matter, we have attempted to classify the patients into a group with vancomycin nephrotoxicity and those without vancomycin nephrotoxicity for a comprehensive comparative evaluation. This decision may be problematic in some cases, especially at the time of patient care, but is significantly facilitated in the current retrospective review. Accordingly, the patients with VNT displayed a distinctive clinical profile as detailed in the Results and the Table 1. In contrast, the reason for the diagnosis of NO-VNT included 

1) AKI long after discontinuation of vancomycin, with a normal serum creatinine at time of this discontinuation; 

2) treatment with oral vancomycin only; 

3) AKI developing before starting vancomycin; 

4) very slow recovery of renal function after discontinuation of vancomycin; 

5) recovery of renal function when still on vancomycin; and 

6) several other episodes of vancomycin treatment not associated with AKI. These findings were not recognized in the VNT group. Furthermore, as detailed in Table 1, more than one reason was noted from the composite criteria for each of the NO-VNT patients. The renal biopsy findings also help to support the diagnosis of NO-VNT. Although renal biopsy changes due to vancomycin nephrotoxicity have not been well defined in literature, over the course of study, a characteristic and quite uniform profile of renal biopsy changes were identified (see below). These findings were not seen in any of the cases of NO-VNT”

Page 7, Lines 10-30 

Page 8, Lines 1-2

5 The reason for inclusion of certain patients in the NO-VNT group is unclear, such as those who developed nephrotoxicity distantly after discontinuation of vancomycin, since they are not necessarily comparable to the patients receiving renal biopsy in suspected VNT cases. Similarly, a patient who received only PO vancomycin, such as case 27, should not be included either since PO vancomycin has minimal systemic absorption. An example of how this may bias findings is how cases 27, 28, and 29 have no evidence of vancomycin casts. If only such patients were included in the NO-VNT group, a misleading conclusion that vancomycin casts are critical to the pathogenesis of VNT could be reached.

 We are grateful for the diagnostic insights of the Reviewers. The cases indeed display a clinico-pathologic profile that is different from that of the typical VNT cases, enabling a tentative diagnosis of NO-VNT. However, this is the main reason they are included. Among different reasons for case assignment/inclusion, as mentioned in many places in the current manuscript, there is a need to not only describe the renal biopsy findings of VNT, but also to assess how specific these changes for the diagnosis of VNT. The renal biopsy findings in these cases would help to answer this question. These cases in our opinion would help us to develop the opinion that there are cases in which treatment of vancomycin is associated with acute kidney injury. In these cases, at least in retrospect, the clinical features tend to negate the possibility of VNT, and the renal biopsy findings would support this critical diagnosis. 

 In relation to the comment “An example of how this may bias findings is how cases 27, 28, and 29 have no evidence of vancomycin casts. If only such patients were included in the NO-VNT group, a misleading conclusion that vancomycin casts are critical to the pathogenesis of VNT could be reached”, we wish to point out the inclusion of these cases reveals vancomycin casts in 6/11 renal biopsies in NO-VNT groups, suggesting that vancomycin cast may serve as an evidence of previous vancomycin exposure /treatment, but may not be critical in the development of vancomycin nephrotoxicity. These conclusions were stated in the original manuscript.

 “ ….Vancomycin casts, sometime quite pronounced, were noted in 6/11 biopsies, casting a doubt on its role as the main cause of VNT” (page 18, last three lines, second paragraph, original manuscript)

“….Findings from the current study suggest that vancomycin casts may add to the renal injury but may not be the main culprit. The findings suggest that ATN and or TIN initially may induce AKI, including single nephron obstruction. This leads to poor proximal handling of vancomycin, formation of uromodulin casts, increased local concentration of vancomycin, precipitation/ crystallization of vancomycin, and localized necrosis and damage of tubular epithelial cells, feed into a self-perpetuating positive feedback loop of local tubular injury, superimposing on the ATN and/or TIN-induced background AKI. This conclusion is also supported by the observation that vancomycin casts were frequently observed in the biopsy from those who were treated with vancomycin but did not develop VNT” (page 19, last sentences, second paragraph, original manuscript). 

Revision: The following sentence is added “Vancomycin casts, sometime quite pronounced, were noted in 6/11 biopsies, casting a doubt on its role as the main cause of VNT. vancomycin cast can serve as an evidence of previous vancomycin exposure /treatment, but may not be critical in the development of vancomycin nephrotoxicity”

Page 20, Lines 8 - 11

6 It is not clear to me how vancomycin casts can be considered a “specific change of VNT” when there was also a significant proportion of patients in the NO-VNT group who also had vancomycin casts on biopsy (particularly if you exclude cases 27, 28, and 29).

 Response: The statement “ Vancomycin tubular casts are another specific change of VNT” in the original manuscript is indeed misleading. We in fact do not mean it, as clearly alluded above. 

 Revision: This sentence is now replaced with another sentence “ Vancomycin tubular casts are an important histopathologic change that are almost always seen in renal biopsies in patients with VNT. However, vancomycin tubular casts are not entirely specific for VNT, since it is also noted in patients in the NO-VNT group.” 

Page 20 – Lines 6-9

7 There is no mention of the limitations of the study, such as selection bias given only patients with more severe or unclear cases of VNT received biopsies. A discussion of the limitations of the study should be included.

 We regret that our enthusiasm has led to a failure to acknowledge the limitations of our study. Indeed, this is also a concern of the Reviewer #2. 

 The following sentences are added before the conclusion “This study is limited by several considerations, which, due to its design and nature, could not have been effectively addressed. These limitations include a single-center design, limiting the generalization of the results; a selection bias with rather small sample size, and inclusion of only cases with severe disease requiring renal biopsies; criteria for diagnostic assignment remain empirical and subjected to interpretation; retrospective nature with limited follow-up duration, as well as cases loss for follow-up. “

Page 22, Lines 17-22

 Minor 

1 In the below sentence from the discussion, was “pathogenetic” was meant to be “pathogenic”? It is not clear how the study would show “pathogenetic” changes for VNT.

“Thus the distinctive pattern of nephrotoxic ATN in the current biopsy implies a specific diagnostic and pathogenetic changes for VNT.”

 Appropriate change is made as recommended by the Reviewer

Page 19, lines 22-23

2 In Table 1, it is not clear what “typical clinical course” means. The description of what a typical clinical course is alluded to within the body of the manuscript, when discussing the rapid improvement, but it is not clearly defined at any point.

 “Typical renal biopsy” is more clearly described in the body of the manuscript given the extensive discussion regarding the biopsy findings, but information in the legend regarding what this means may be helpful.

 The word “typical” may not be the best choice.

 An entire section in results (page 10, lines 13-30, page 11, lines 1-13) and discussion (page 16, lines 1-30) is devoted to this “characteristic” clinical course in discussion. We hope that the message is adequately delivered. 

 Additional notations for the figure legends are now added.

Page 25 – Lines 25-27, Page 26 – lines 1 and 15

3 In Table 1, there is only a single vancomycin dose and vancomycin trough level listed per case. The dose of vancomycin may change based on the trough levels and trough levels are expected to vary. Were only the maximum dose and trough level reported?

 As pointed out by the Reviewer, variation of vancomycin dose over the treatment course is indeed noted in some cases and the vancomycin blood levels were variable over time. However, we have included only the maximal dose and the corresponding blood levels, to avoid overcrowding an already quite busy list of data. This also reflects the idea that these values are perhaps best correlate with the development of vancomycin nephrotoxicity. 

4 How was baseline creatinine defined?

 Baseline serum creatinine is defined as the lowest level before administration of vancomycin. This value is either recorded during a previous hospital admission for reasons unrelated to the current acute kidney injury, or during the current admission, before the development of acute kidney injury. 

Revision: The following sentence is added to the revised manuscript “Baseline serum creatinine is defined as the lowest level before the administration of vancomycin. This value is either recorded during a previous hospital admission for reasons unrelated to the current acute kidney, or during the current admission, before the development of acute kidney injury”

Page 7; Lines 9-12

5 In Table 2, it is not clear what “A>>>C” means. “A+C” presumably means both acute and chronic injury is present, but this is also not entirely clear.

 Response and Revision: Pertinent explanatory foot notes are now added to the Table 2

Page 38 – Lines 1-7 in Notes

6 The below sentence from the discussion, is phrased in a manner that is difficult to understand.

“Thus, when severe infection supervenes, vancomycin, as the antibiotic of choice, is given in standard dose, upon which severe AKI occurs, reflected by a highly elevated serum creatinine, develops quickly sometime as early as one day after treatment.”

 Response and Revision: This sentence is indeed so dense that we now have some difficulty to grasp it.

 The sentence is revised as follows “Thus, when severe infection supervenes, vancomycin, as the antibiotic of choice, is given in standard dose. This treatment is followed by severe AKI, with a marked and quick elevation of serum creatinine from a normal base line, which develops quickly, sometime as early as one day after treatment.”

Page 16; Lines 9-12

 There are several sentences that end with commas rather than periods, which should be corrected.

 Response and Revision: We are sorry for the oversight. These are corrected. 

 Reviewer 2 Comments Response and Revision

1 The first limitation of the study is the single center design, which could decrease the generalizability of the results and their external validity, as well as its retrospective nature. Moreover, a further limitation of the study is represented by the small sample of patients. 

 Response and Revision: We are grateful for the insights. Indeed, this concern is also shared by the Reviewer #1. The following sentences are now added to the revised manuscript. 

 The following sentences are added before the conclusion “This study is limited by several considerations, which, due to its design and nature, could not have been effectively addressed. These limitations include a single-center design, limiting the generalization of the results; a selection bias with rather small sample size, and inclusion of only cases with severe disease requiring renal biopsies; criteria for diagnostic assignment remain empirical and subjected to interpretation; retrospective nature with limited follow-up duration, as well as cases loss for follow-up.

Page 22; Lines 17-22

 We trust we have adequately addressed all comments and concerned from the Reviewers and the Editor, and have made appropriate revisions as shown in the revised manuscript. We hope the manuscript in its current form meet the high standard of publication of PLOS One.

Yours sincerely

Dr. Rajesh Nachiappa Ganesh

---

## [Decision Letter · Decision Letter 1]

16 Nov 2023

Vancomycin Nephrotoxicity: A Comprehensive Clinico-pathological Study

PONE-D-23-17096R1

Dear Dr. Nachiappa Ganesh,

We’re pleased to inform you that your manuscript has been judged scientifically suitable for publication and will be formally accepted for publication once it meets all outstanding technical requirements.

Kind regards,

Fabio Sallustio, PhD

Academic Editor

PLOS ONE

Additional Editor Comments (optional):

Reviewers' comments:

Reviewer's Responses to Questions

**Comments to the Author**

1. If the authors have adequately addressed your comments raised in a previous round of review and you feel that this manuscript is now acceptable for publication, you may indicate that here to bypass the “Comments to the Author” section, enter your conflict of interest statement in the “Confidential to Editor” section, and submit your "Accept" recommendation.

Reviewer #1: All comments have been addressed

2. Is the manuscript technically sound, and do the data support the conclusions?

Reviewer #1: (No Response)

3. Has the statistical analysis been performed appropriately and rigorously? 

Reviewer #1: (No Response)

4. Have the authors made all data underlying the findings in their manuscript fully available?

Reviewer #1: (No Response)

5. Is the manuscript presented in an intelligible fashion and written in standard English?

Reviewer #1: (No Response)

6. Review Comments to the Author

Reviewer #1: (No Response)

7. PLOS authors have the option to publish the peer review history of their article (what does this mean?). If published, this will include your full peer review and any attached files.

Reviewer #1: No

---

## [Editor Report · Acceptance letter]

20 Nov 2023

PONE-D-23-17096R1 

Vancomycin Nephrotoxicity, A Comprehensive Clinicopathological Study 

Dear Dr. Nachiappa Ganesh:

I'm pleased to inform you that your manuscript has been deemed suitable for publication in PLOS ONE. Congratulations! Your manuscript is now with our production department. 

Kind regards, 

on behalf of

Dr. Fabio Sallustio 

Academic Editor

PLOS ONE